# Shared Memory for Multi-agent Lifelong Pathfinding

## Abstract

Multi-agent reinforcement learning (MARL) demonstrates significant progress in solving cooperative and competitive multi-agent problems in various environments. One of the principal challenges in MARL is the need for explicit prediction of the agents' behavior to achieve cooperation. To resolve this issue, we propose the Shared Recurrent Memory Transformer (SRMT) which extends memory transformers to multi-agent settings by pooling and globally broadcasting individual working memories, enabling agents to exchange information implicitly and coordinate their actions. We evaluate SRMT on the Partially Observable Multi-Agent Pathfinding problem in a toy Bottleneck navigation task that requires agents to pass through a narrow corridor and on a POGEMA benchmark set of tasks. In the Bottleneck task, SRMT consistently outperforms a variety of reinforcement learning baselines, especially under sparse rewards, and generalizes effectively to longer corridors than those seen during training. On POGEMA maps, including Mazes, Random, and MovingAI, SRMT is competitive with recent MARL, hybrid, and planning-based algorithms. These results suggest that incorporating shared recurrent memory into the transformer-based architectures can enhance coordination in decentralized multi-agent systems.

## 1 Introduction

Multi-agent systems have significant potential to solve complex problems through distributed intelligence and collaboration. However, coordinating the interactions between multiple agents remains challenging, often requiring sophisticated communication protocols and decision-making mechanisms. We propose a novel approach to address this challenge by introducing a *shared memory* as a global workspace for agents to coordinate behavior. The global workspace theory (Baars, 1988) suggests that in the brain, there are independent functional modules that can cooperate by broadcasting information through a global workspace. Inspired by this theory, we consider the agents in Multi-Agent Pathfinding (MAPF) as independent modules with shared memory and propose a *Shared Recurrent Memory Transformer* (SRMT) as a mechanism for exchanging information to improve coordination and avoid deadlocks. SRMT extends memory transformers (Burtsev et al., 2020; Bulatov et al., 2022; Yang et al., 2022; Cherepanov et al., 2024) to multi-agent settings by pooling and globally broadcasting individual working memories.

In this study, we test the shared recurrent memory approach on Partially Observable Multi-agent Pathfinding (PO-MAPF) (Stern et al., 2019), where each agent aims to reach its goal while observing the state of the environment, including locations and actions of the other agents and static obstacles, only locally. Multi-agent pathfinding can be considered in the *decentralized* manner, where each agent independently collects rewards and makes decisions on its actions. There are many attempts to solve this problem: in robotics (Van den Berg et al., 2008; Zhu et al., 2022), in machine and reinforcement learning field (Damani et al., 2021; Ma et al., 2021b; Wang et al., 2023; Sartoretti et al., 2019; Riviere et al., 2020). Such methods often involve manual reward-shaping and external demonstrations. Also, several works utilize the communication between agents to solve decentralized MAPF (Ma et al., 2021a; Li et al., 2022; Wang et al., 2023). However, the resulting solutions are prone to deadlocks and poorly generalizable to the maps not used for training. In this work, we compare SRMT to a range of reinforcement learning baselines and show that it consistently outperforms them in the Bottleneck navigation task. Tests on POGEMA benchmark Skrynnik et al.

(2024a) show that SRMT is competitive with numerous recent MARL, hybrid, and planning-based algorithms.

## 2 RELATED WORK

### 2.1 SHARED MEMORY AND COMMUNICATION IN MULTI-AGENT REINFORCEMENT LEARNING

Shared memory is designed to help agents coordinate their behavior, and it is closely related to existing approaches in multi-agent reinforcement learning (MARL), particularly those involving inter-agent communication. Compared to a single-agent reinforcement learning, providing agents with communication protocol presents a significant challenge in MARL (Foerster et al., 2016; Iqbal & Sha, 2018; Zhang et al., 2018). Common strategies to address this problem include (1) a centralized setting, where a central controller aggregates information from all agents; (2) a fully decentralized setting, where agents make decisions based solely on local observations; and (3) a decentralized setting with networked agents, allowing agents to share local information with each other (Zhang et al., 2021; Hu et al., 2023; Nayak et al., 2023; Agarwal et al., 2019).

In multi-agent pathfinding (MAPF) with reinforcement learning, various methods fit within these three categories. Decentralized methods without communication include approaches such as IQL (Tan, 1993), VDN (Sunehag et al., 2018), QMIX (Rashid et al., 2020), QPLEX (Wang et al., 2021), Follower (Skrynnik et al., 2024b), and MATS-LP (Skrynnik et al., 2024c). These methods propose to implement the agent decision-making based only on local information. Notably, VDN, QMIX, and QPLEX adopt centralized training with a joint Q-function but operate in a decentralized manner during execution with individual Q-functions. In contrast, centralized methods such as LaCAM (Okumura, 2023) and RHCR (Li et al., 2021) employ a centralized search-based planner. Finally, decentralized methods with communication, such as DCC (Ma et al., 2021b), MAMBA (Egorov & Shpilman, 2022), and SCRIMP (Wang et al., 2023), allow agents to share information to enhance coordination and avoid collisions. These methods utilize various communication strategies, including selective information sharing (DCC), discrete communication protocols (MAMBA), and scalable communication mechanisms based on transformer architectures (SCRIMP), all aimed at improving agent cooperation in complex and dynamic environments.

MAMBA (Egorov & Shpilman, 2022) is a pure MARL approach that uses communication and centralized training within a Model-Based Reinforcement Learning framework, featuring discrete communication and decentralized execution. A 3-layer transformer serves as the communication block with its outputs used by the agents to update their world models and make action predictions. Each agent maintains its own version of the world model, which can be updated through the communication block.

QPLEX (Wang et al., 2021) is a pure multi-agent reinforcement learning method that employs multi-agent Q-learning with centralized end-to-end training, providing inter-agent communication. QPLEX learns to factorize a joint action-value function to enable decentralized execution. The approach uses a duplex dueling mechanism that connects local and joint action-value functions, allowing agents to make independent decisions while benefiting from centralized training.

Follower (Skrynnik et al., 2024b) is a learnable method for lifelong MAPF without communication that uses a centralized path planning with a modified A* heuristic search to reduce agents' collisions. MATS-LP (Skrynnik et al., 2024c) is also a learnable method for lifelong MAPF without communication that uses a learnable policy combined with Monte Carlo Tree Search (MCTS) to reason about possible future states.

RHCR (Li et al., 2021) is a purely search-based centralized planner that does not require training. It decomposes the lifelong MAPF into a sequence of windowed MAPF instances, with re-planning occurring according to a predetermined schedule, resolving collisions within the current planning window only.

The proposed *shared recurrent memory* approach fits into a decentralized setting with networked agents. Unlike other methods, SRMT allows agents to communicate indirectly through a shared memory. Each agent uses a recurrently updated memory and learns to read and write its individual memory representations to a shared space. This allows agents to retain information over time steps in the episode and enables the effective individual and collective decision-making process in

pathfinding tasks. Furthermore, unlike approaches that rely on a joint Q-function for centralized training, our method maintains decentralization throughout training and execution. The fully decentralized setting of the multi-agent system might be preferable over the centralized one in many real-world applications.

## 2.2 Shared memory and memory transformers

SRMT extends the memory transformers (Burtsev et al., 2020; Bulatov et al., 2022; Yang et al., 2022; Cherepanov et al., 2024) to multi-agent settings by pooling and globally broadcasting individual memories of each agent.

Memory Transformer (Burtsev et al., 2020) augments the standard Transformer architecture (Vaswani et al., 2017) by introducing special memory tokens appended to the input sequence, providing the additional operational space for the model. These memory tokens are trainable and are used by the model as working memory. In the Recurrent Memory Transformer (RMT) (Bulatov et al., 2022), memory tokens transfer information between segments of a long input sequence. In this case, multiple memory tokens act as a recurrent state, effectively turning the transformer into a recurrent cell that processes each segment as input at each time step.

Agent Transformer Memory (ATM) (Yang et al., 2022) introduces a transformer-based working memory into multi-agent reinforcement learning. Each ATM agent maintains a memory buffer of the last N memory states, sequentially updated by a transformer network. This approach is similar to the RMT, but instead of using only the latest memory state, ATM uses the several most recent memory states. Additionally, each memory state in ATM consists of a single vector, whereas in RMT, multiple memory vectors work together as a recurrent hidden state.

Recurrent Action Transformer with Memory (RATE) (Cherepanov et al., 2024) extends the Decision Transformer (Chen et al., 2021) by incorporating memory tokens and a dedicated memory update module, the Memory Retention Valve, which updates memory states to effectively handle long segmented trajectories.

Relational Recurrent Neural Network (RRNN) (Santoro et al., 2018) utilizes the multi-head dot product attention to update the memory state given the new input data. Then it employs the modification of a standard LSTM (Hochreiter, 1997) treating the memory matrix as a matrix of recurrent cell states to predict the model outputs.

Approaches such as ATM, RATE, and RRNN are focused on maintaining individual memory states for each agent. In contrast, SRMT extends recurrent memory to multi-agent RL and facilitates the *shared* access to individual agents' memories, enabling coordination and joint decision-making among all agents in the environment.

## 3 Shared Recurrent Memory Transformer

Multi-agent pathfinding task is set as follows. The agents interact in the two-dimensional environment represented as graph $G = (V, E)$ with the vertices corresponding to the locations and the edges to the transitions between these locations. The timeline consists of discrete time steps. The predefined final step of the agent interaction episode is called the episode length. At the beginning of the episode, each agent $i$ is given a start location $s_i \in V$ and a goal location $g_i \in V$ to be reached until the end of the episode. At each time step, an agent performs an action to stay in its current location or move to an adjacent one. The task of multi-agent pathfinding is to make each agent reach its goal until the end of the episode without colliding with the other agents.

In this study, we work with a decentralized Partially Observable Multi-agent Pathfinding (PO-MAPF) (Stern et al., 2019). Decentralization of MAPF means that the multi-agent system has no global controller, each agent performs actions and collects rewards independently of the others. Also, each agent observes the environment obstacles, other agents, and their goals only locally, from a squared window of fixed size centered at the agent's current location. Formally, the partially observable multi-agent Markov decision process $M$ is defined (Bernstein et al., 2002):

$$M = \langle S, U, A, P, R, O, \mathcal{O}, \gamma \rangle,$$

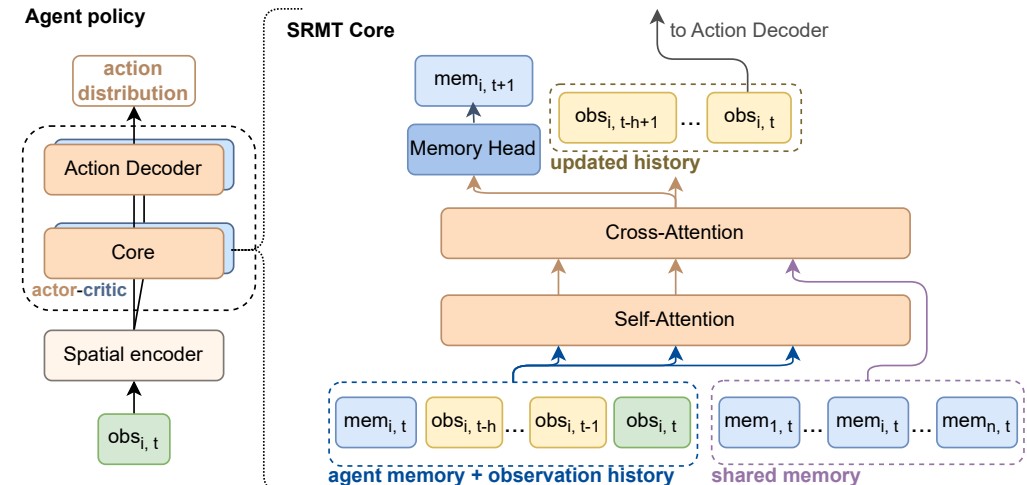

Figure 1: **Shared Recurrent Memory Transformer architecture.** SRMT pools recurrent memories $mem_{i,t}$ of individual agents at a moment $t$ and provides global access to them via cross-attention.

where $S$ is the set of environment states, $U = \{1, \ldots, n\}$ represents the agents, $A$ is the set of possible actions with action of agent $u$ denoted as $a^{(u)}$. At each time step of the episode, agents form a joint action $a^{(U)} \in A^n$. Then, the environment state is updated with the transition function

$$P(s' \mid s, a^{(U)}) : S \times A^n \times S \to [0, 1].$$

After all agents performed their actions, each of them receives the following scalar reward

$$R(s, u, a^{(U)}) : S \times U \times A^n \to \mathbb{R},$$

reflecting the agent's performance during the previous step. The future rewards are discounted by a factor $0 \leq \gamma \leq 1$ defining their importance. Before the next step, each agent also receives its local observation $o^{(u)} \in O$ based on the following global observation function

$$\mathcal{O}(s, a) : S \times A \to O.$$

The historical sequence of observations and respective actions $h^{(u)}$ is preserved for each agent to estimate a learnable stochastic policy

$$\pi^{(u)}(a^{(u)} \mid h^{(u)}) : T \times A \to [0, 1].$$

To train the policy approximation model the expected cumulative reward is used as an objective function to maximize over time.

We approximate the policy with a deep neural network depicted in Figure 1. The policy consists of the spatial encoder to process agent observation and the actor-critic network. The agents are assumed to be homogeneous, so the policy is shared between agents during training. The spatial encoder uses ResNet (He et al., 2016) with an additional Multi-Layer Perceptron (MLP) in the output layer. The Action Decoder and the Critic Head are Dense layers. The policy neural network is presented in Figure 1. We adopt the Skrynnik et al. (2024b) approach for configuring the policy-approximating model and input-output data pipelines. The model input is the agent's local observation tensor of shape $3 \times m \times m$ tensor, where $m$ is the observation range. The channels of the tensor encode the static obstacle locations combined with the current path, the other agents, and their targets.

SRMT is used as a core subnetwork of the actor-critic model. As the main mechanism in SRMT, we consider the attention block implemented in the Huggingface GPT-2[1] model. In SRMT, the input sequence for each agent at each time step is constructed by combining three key components: the

---

[1]https://huggingface.co/docs/transformers/model_doc/gpt2

agent's personal memory vector; the historical sequence of the agent's observations from the past $\hat{h} = 8$ time steps; and the current step observation. This sequence undergoes a full self-attention mechanism. Next, the output of the self-attention is passed through a cross-attention layer between current hidden representations and shared memory. The shared memory consists of a globally accessible, ordered sequence of all agents' memory vectors for the current time step. This interaction between personal and shared memory enables each agent to incorporate global context into the decision-making process. The resulting output is then passed through a memory head, which updates the agent's personal memory vector for the next time step. Simultaneously, the model output is also fed into the decoder part of the policy model, which generates the agent's action.

## 4 EXPERIMENTS AND RESULTS

### 4.1 CLASSICAL MAPF ON BOTTLENECKS

We use the POGEMA (Skrynnik et al., 2024a) framework for our experiments. In POGEMA the two-dimensional environment is represented as a grid composed of obstacles and free cells. At each time step each agent can perform two types of actions: moving to an adjacent cell or remaining at their current position. Agents have limited observational capabilities, perceiving other agents only within a local $R \times R$ area centered on their current position. The episode ends when the predefined time step, episode length, is reached. The episode can also end before this time step if certain conditions are met, i.e. all agents reach their goal locations.

As an initial test of our approach, we selected a simple two-agent coordination task where the agents must navigate through a narrow passage. The environment consists of two rooms connected by a one-cell-wide corridor, as illustrated in Fig. 2a. Each agent has a $5 \times 5$ field of view and starts in one room, and their goal is located in the opposite room, requiring both agents to pass through the narrow corridor to complete the task. To train the model, we utilized 16 maps with corridor lengths uniformly selected between 3 and 30 cells. To evaluate the performance, we utilize three metrics:

- *Cooperative Success Rate (CSR)* – a binary measure indicating whether all agents reached their goals before the episode's end;

- *Individual Success Rate (ISR)* – the fraction of the agents that achieved their goals during the episode;

- *Sum-of-Costs (SoC)* – the total number of time steps taken by all agents to reach their respective goals (the lower the value the better).

The policy function is approximated with a deep neural network (see Fig. 1, left). To benchmark the effectiveness of SRMT, we compared it against MAMBA, QPLEX, ATM, RATE, and RRNN models. As additional baselines that also serve as ablations of SRMT, we evaluate the following core subnetwork architectures: the recurrent memory transformer that allows agents to process individual memory representations without sharing them (RMT), the memoryless transformer (Attention), the empty core that uses the direct connection of spatial encoder to the actor-critic action decoder (Empty), and GRU RNN (RNN) to assess the difference between the attention-based and RNN-based observation processing. Additional details regarding the training procedure can be found in Appendix A.1. For the Bottleneck task, we apply no advanced heuristics or methods for path planning to test the impact of memory addition. Our path-planning strategy is simple: each agent aims to follow the shortest path to the goal at each time step. If according to the planned movements, the agents may collide, their final decision will be to retain their current positions until the next step. We hypothesize that a shared memory mechanism will help to solve such bottleneck problems.

We first evaluated SRMT against the baseline models using three variations of the reward function: *Directional*, *Moving Negative*, and *Sparse*.[2] In the Directional setting, the agent was rewarded for reaching the goal and for every step that brought it closer to the goal. In the Moving Negative strategy movements are slightly penalized to minimize the path to the goal but no information about goal location is provided. In contrast, the Sparse reward function only gives a reward when the agent successfully achieves the goal cell, offering no intermediate rewards.

---

[2]See Appendix A for details of rewards and extra results.

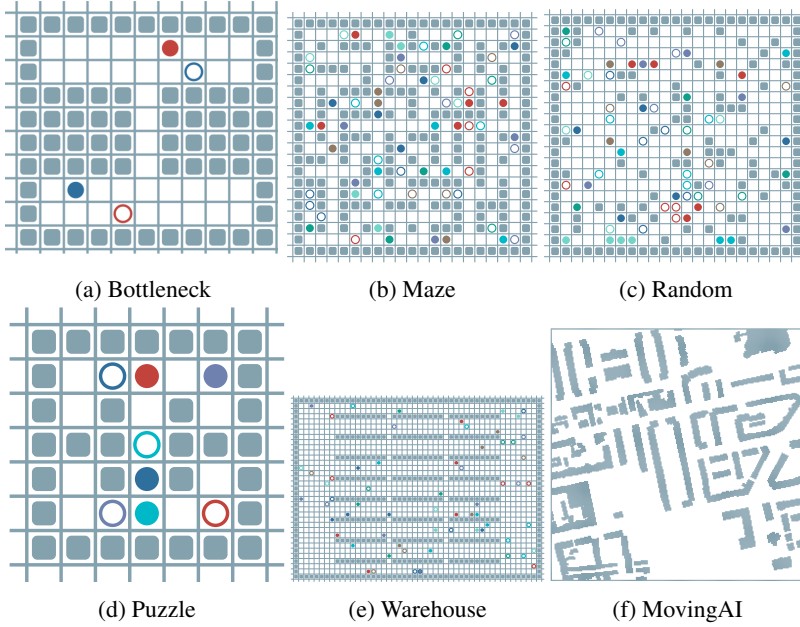

(a) Bottleneck       (b) Maze       (c) Random

(d) Puzzle       (e) Warehouse       (f) MovingAI

Figure 2: **Examples of environments.** (a) Bottleneck task. This is a toy task on coordination. Two agents start in rooms opposite their goals and should coordinate passing the corridor. Agents are shown as solid-colored circles, their goals are empty circles with the same border color. (b)-(f) Maps from POGEMA benchmark (images for POGEMA maps are from (Skrynnik et al., 2024a)). POGEMA allows testing the planning methods' generalization across different maps and problem sizes.

The comparison of evaluation scores, shown in Figure 3, demonstrates that SRMT successfully solves the task under considered reward functions. In particular, SRMT consistently performed well, even in the challenging Moving Negative and Sparse reward scenarios where other methods struggled. The ability to coordinate agents via shared memory proved critical, especially when feedback from the environment was minimal.

Among the non-memory sharing methods, RMT achieved the best performance, outperforming ATM, RATE, RNN, GRU-based RNN, and the Attention models. However, without the shared memory, RMT's performance in the Sparse reward setting was still limited compared to SRMT, highlighting the importance of the global information exchange.

To further assess the generalization capabilities of the trained policies, we evaluated them on bottleneck environments with corridor lengths significantly larger than those used during training, ranging from 5 to 1000 cells. The evaluation results[3] for the Sparse and Moving Negative reward functions are shown in Figure 4. While implementing the RRNN, RATE, and ATM, we noticed the common feature of these architectures to initialize the memory state with some pre-defined values (unit vector or random vector sampled from the Normal distribution with zero mean and unit variance). In contrast, the SRMT memory state is initialized with the values generated on the first step of the episode. To further explain the drastic difference in CSR scores of baseline memory approaches (RRNN, RATE, ATM) and SRMT, in Figure 4 we provided the results of an additional experiment: we modified the initial state of the RATE memory to be generated from the initial observation of the agent similar to SRMT. The resulting performance with the Moving Negative rewarding scheme (RATE_gen) significantly increased compared to the original RATE implementation showing the importance of the proper memory initialization procedure.

The results indicate that trained with the Sparse reward function, the SRMT-based policy consistently outperforms the baseline methods from the related works and the SRMT ablations regarding

---

[3]For these evaluations, we adjusted the episode length to $2 \cdot corridor\_len + 100$ to ensure that both agents had sufficient time to reach their goals within each episode.

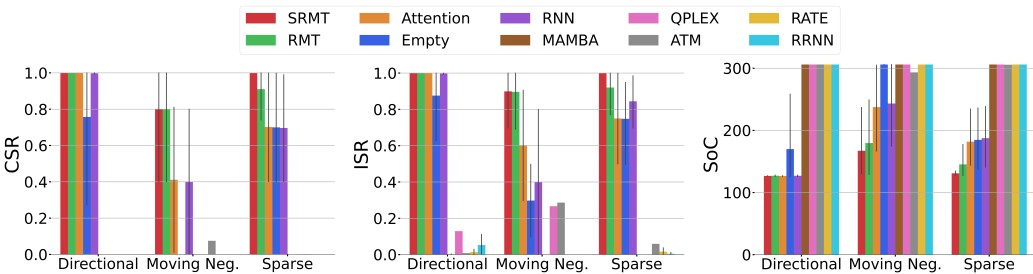

Figure 3: **SRMT effectively solves the Bottleneck Task with different reward functions.** Trained with Directional (positive when moved towards a goal and achieved it) reward, SRMT clearly outperforms the communication (MAMBA, QPLEX) and memory (ATM, RATE, RRNN) baselines. The RMT, Attention, and RNN ablations also solve the task. For the case with the negative reward for movement and no directional reward (Moving Negative) SRMT and RMT without shared memory demonstrate the clear advantage over the memory-less ablations of SRMT (Attention, Empty, RNN) and the communicative and memory baselines (MAMBA, QPLEX, ATM, RATE, RRNN). With the Sparse (on-goal only) reward, SRMT maintains the score while other methods drop. Error bars indicate 95% confidence intervals. For CSR and ISR higher values are better, for SoC – the lower the better.

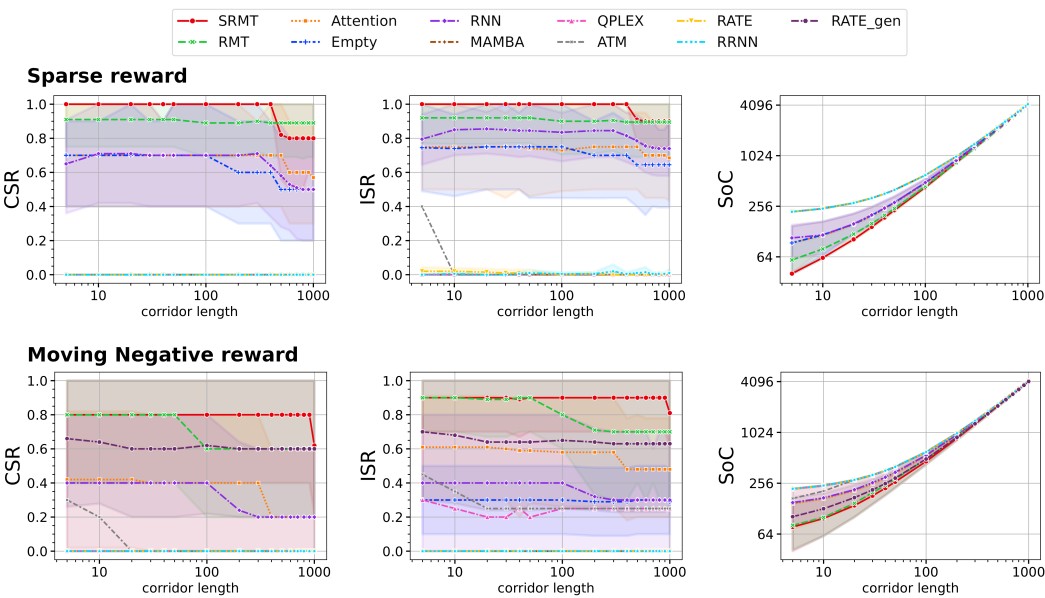

Figure 4: **SRMT agents generalize on corridor lengths up to 1000.** After training on corridor sizes from 3 to 30 cells all methods were evaluated on longer passages up to 1000. All non-zero performing models show good scaling up to the corridor length of 100. For the Sparse reward, SRMT leads up to 400 and then drops below RMT for collective performance. For the Moving Negative reward, SRMT shows the top-1 performance on all three metrics. The shaded area indicates 95% confidence intervals.

both the individual success rate (ISR) and the total time spent in the environment (SoC). Remarkably, SRMT scales effectively up to corridor lengths of 400 cells without any significant performance degradation. For corridor lengths beyond 400 steps, the Cooperative Success Rate (CSR) of SRMT drops from 1.0 to 0.8, which surpasses other models, except for RMT. For the Moving Negative reward function, resulting policies with SRMT show the top-1 scores for any corridor length up to 1000 (see Fig. 4). Evaluation scores for the other reward functions and analysis of memory representations can be found in Appendix A.

## 4.2 LIFELONG MAPF

Lifelong multi-agent pathfinding (LMAPF) is an extension of MAPF, where agents receive new destinations upon completing their current goals. The main quality metric for this task setting is the *average throughput* calculated as the average number of goals reached by all agents per episode step.

To train SRMT in the LMAPF setting we use the set of 40 maze-like environments (Fig. 2b) of size $65 \times 65$ following the (Skrynnik et al., 2024b) training procedure. The same architecture as depicted in Fig. 1 is used for the policy approximation model with a larger number of layers compared to the classical MAPF on bottleneck environments. The detailed listing of hyperparameters can be found in the Appendix A.1 Table 1. We consider the following reward strategy for lifelong SRMT experiments: if the agent follows the planned path to the goal, it receives a small positive reward $r = 0.01$. Otherwise $r = 0$. To define the path from the current location to the goal, we consider the A* shortest path algorithm and more advanced Heuristic Path Decider method presented in the Follower (Skrynnik et al., 2024b). A* plans the shortest individual path to the goal, while heuristic search finds evenly dispersed paths to alleviate the congestion problems in environments with large populations of agents.

To assess the effectiveness of SRMT for LMAPF, we compare it with LMAPF baselines from the POGEMA benchmark (see Fig. 5). We trained the SRMT with 64 agents for 1B environment steps,

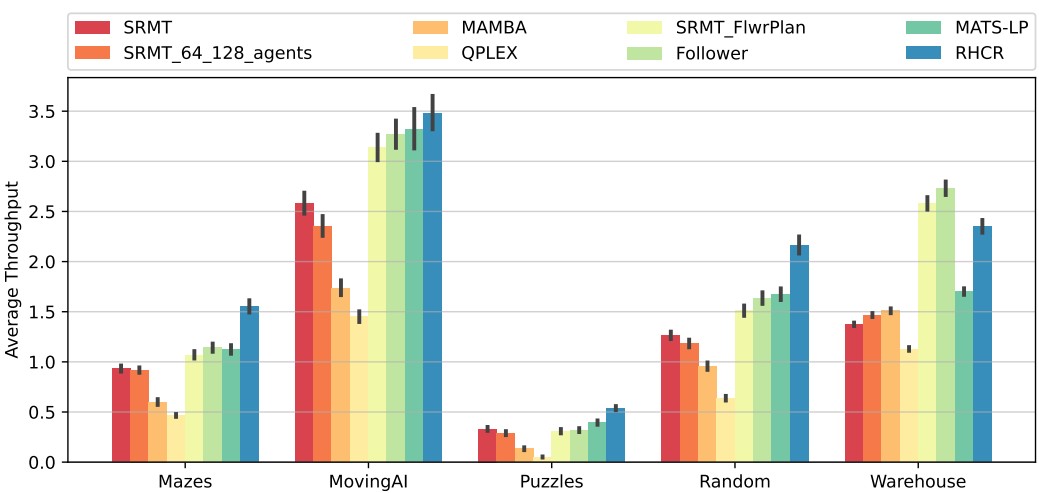

Figure 5: **SRMT outperforms other MARL methods in different environments.** SRMT trained on Mazes shows robust generalization when evaluated on maps not seen during training. SRMT outperforms MARL baselines MAMBA and QPLEX on all maps except the Warehouse environment. Mixed training with 64 or 128 agents (SRMT 64-128) does not affect the generalization abilities of the method. In the Warehouse environment, the average throughput of SRMT with a reward function based on the Follower heuristic path search (SRMT-FlwrPlan) surpasses that of MAMBA, MATS-LP, QPLEX, and RHCR methods. Error bars indicate 95% confidence intervals.

SRMT (SRMT 64-128) with the mixture of 64 and 128 agents for 400M steps, and SRMT with planning algorithm from the Follower (SRMT-FlwrPlan) with 64 agents for 600M environment steps.

Figure 5 presents the average throughput of SRMT and the baseline methods across different types of environments. The results show that SRMT trained on Mazes generalizes robustly to unseen maps, outperforming MARL baselines MAMBA and QPLEX on all maps except the Warehouse environment. The mixed training with 64 and 128 agents (SRMT 64-128) did not significantly affect the generalization abilities of the method. In Appendix A.2 Figure 10, we also provided detailed scores for evaluations of MovingAI task with increasing numbers of agents to show the scalability of SRMT.

In the Warehouse environment, characterized by narrow corridors and high congestion, the average throughput of SRMT with a reward function based on the Follower's heuristic path search (SRMT-

FlwrPlan) surpasses that of MAMBA, MATS-LP, QPLEX, and even the centralized planning method RHCR. This indicates that integrating heuristic planning into the SRMT framework enhances performance in highly congested settings.

To further evaluate the performance of SRMT, we measured its effectiveness using high-level MAPF metrics from the POGEMA benchmark (Skrynnik et al., 2024a). These metrics provide an assessment of different aspects of multi-agent coordination and scalability:

- *Performance* shows the relation of the average throughput of each method to the best value among the considered methods. The metric is averaged over the results obtained on the Random ($20 \times 20$ grids with randomly placed obstacles) and Mazes (maze-like grids of size $21 \times 21$) sets of maps

- *Pathfinding* is a binary value that demonstrates if the path of a single agent on a large map is optimal. To measure it, the methods were evaluated on $256 \times 256$ city maps from the MovingAI (Stern et al., 2019) benchmark.

- *Congestion* is calculated as the average density of the agents present in the observations of each agent compared to the overall density of the agents on the Warehouse map.

- *Cooperation* metric is dedicated to the ability of the method to resolve complex situations. It is obtained as the relative average throughput with respect to the best-performing classical LMAPF solvers measured on a small set of hand-crafted $5 \times 5$ Puzzles maps. The maps were constructed to contain difficult patterns requiring agents' cooperation.

- *Out-of-Distribution* demonstrates the Performance metric values on the maps unseen during training. For evaluation, the MovingAI-tiles $64 \times 64$ patches of MovingAI maps were used.

- *Scalability* describes how the runtime of each method changes with the growing number of agents on the Warehouse map. The metric is calculated as the ratio between the relative runtimes and relative numbers of agents.

Figure 6 presents the results of a comparison of SRMT and the other methods across these key metrics. The bar chart illustrates that SRMT and its variants demonstrate competitive performance, particularly in Scalability and Pathfinding. When integrated with the Follower planning, SRMT performs best in Congestion management. The centralized planning method RHCR leads in several metrics, notably Cooperation, Out-of-distribution, Performance, and Pathfinding, reaching nearly 100% in these categories, but is performing worse in Scalability. MAMBA shows strong performance in Congestion management and Scalability.

In terms of *Performance*, SRMT achieves high average throughput on Random, Mazes, and MovingAI environments, indicating effective goal-reaching behavior in diverse settings. For the *Pathfinding* metric, SRMT and its variants perform exceptionally well, suggesting that the agents find near-optimal paths even on large-scale maps.

Regarding *Congestion*, MAMBA slightly outperforms SRMT, showcasing better handling of high-density agent scenarios on the Warehouse map. However, SRMT maintains competitive performance, demonstrating its ability to navigate congested areas effectively. Furthermore, SRMT performs best in Congestion management when integrated with Follower planning.

The *Cooperation* metric highlights the centralized method RHCR's superiority in resolving complex situations requiring intricate coordination. While SRMT does not reach RHCR's level in this aspect, it outperforms the other decentralized methods except MATS-LP with MCTS planning, indicating reasonable cooperative capabilities without centralized control.

For *Out-of-Distribution* generalization, SRMT variations exhibit strong performance, maintaining high throughput on unseen maps. This emphasizes SRMT's robustness and ability to generalize beyond the training environments.

Our experiments demonstrate that incorporating shared memory into transformer-based architectures significantly enhances coordination in decentralized multi-agent systems. SRMT effectively enables agents to implicitly share information, enabling better decision-making and conflict avoidance. The results indicate that SRMT not only outperforms traditional MARL baselines in complex environments but also generalizes well to unseen maps and scales efficiently with an increasing number of agents.

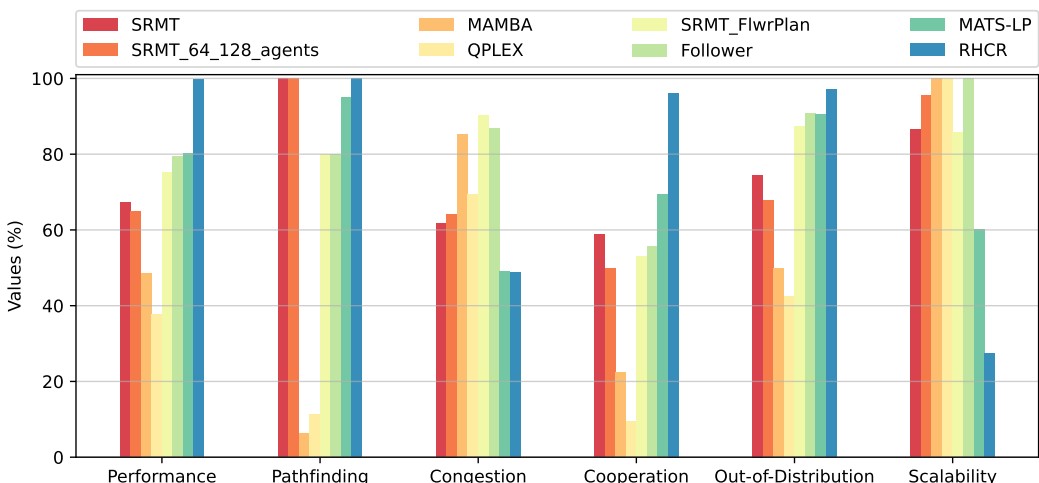

Figure 6: **Comparison of SRMT and other methods across key performance metrics in multi-agent pathfinding.** The bar chart compares the performance of SRMT and its variants (SRMT 64-128, SRMT-FlwrPlan) against other methods – MAMBA, QPLEX, Follower, MATS-LP, and RHCR – across six metrics: Performance, Pathfinding, Congestion, Cooperation, Out-of-Distribution, and Scalability. SRMT and its variants demonstrate competitive performance, particularly in Scalability and Pathfinding. When integrated with Follower planning, SRMT performs best in Congestion management. The centralized planning method RHCR leads in several metrics, notably Cooperation, Out-of-distribution, Performance, and Pathfinding, reaching nearly 100%. MAMBA shows strong performance in Congestion management and Scalability.

In highly congested environments like the Warehouse, integrating heuristic planning into the SRMT framework (SRMT-FlwrPlan) further improves performance, suggesting that combining learning-based approaches with planning algorithms can be beneficial in certain settings. The competitive performance of SRMT across various high-level metrics underscores its potential as a scalable and robust solution for complex multi-agent pathfinding problems in decentralized settings, where explicit communication or centralized control may not be feasible.

## 5 CONCLUSION

In this paper, we introduced a novel Shared Recurrent Memory Transformer (SRMT) architecture for enhancing coordination in multi-agent systems. SRMT enables agents to exchange information implicitly and coordinate actions without explicit communication protocols. Our experimental results on the bottleneck navigation task demonstrate that SRMT consistently outperforms baseline models, especially in challenging scenarios with sparse rewards and extended corridor lengths. The shared memory mechanism allows agents to generalize their learned policies to environments with significantly longer corridors than those seen during training, demonstrating the scalability and robustness of our approach. On POGEMA maps including Mazes, Random, Moving-AI, and Warehouse SRMT is competitive with various recent MARL, hybrid, and planning-based algorithms. These findings highlight the potential of incorporating shared memory structures in transformer-based architectures for multi-agent reinforcement learning.

### LIMITATIONS

As in the majority of research related to Multi-Agent Pathfinding (MAPF), in this work, we assume that the agents have flawless localization and mapping abilities. Our primary focus is on the decision-making aspect of the problem. We also consider that the agents execute actions accurately and that their moves are synchronized. Additionally, we treat obstacles as fixed elements of the environment.

Finally, it is important to note that our approach, like other prominent learnable methods designed for (PO)-MAPF – such as PRIMAL (Sartoretti et al., 2019), PRIMAL2 (Damani et al., 2021), DHC (Ma et al., 2021a), and PICO (Li et al., 2022) – does not offer theoretical guarantees that agents will reach their destinations. However, extensive experimental evidence from our work and the referenced studies, demonstrates that these learnable methods are practically powerful and scalable solutions for complex MAPF problems.

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

## A APPENDIX

### A.1 TRAINING DETAILS

The environments were created with POGEMA Skrynnik et al. (2024a) framework. The Sample Factory codebase (Petrenko et al., 2020) was used for policy model training.

Training parameters for all tested methods are listed in Table 1. A single Tesla P100 was used for training policy models for approximately 1 hour each. The results for models trained with Sparse and Dense reward functions were averaged over 10 runs with different random seeds. The results of training policies with Directional and Directional Negative rewards were averaged over 5 runs with different random seeds as they showed less variation during training. Each run evaluation was first averaged over 10 different evaluation procedure random seeds. We carried out the grid search for the SRMT training entropy coefficient (range $[0.00001, 0.0003]$) and learning rate (range $[0.01, 0.05]$).

Table 1: Models configuration and training hyperparameters.

| Parameter | MAPF (all models) | SRMT LMAPF |
|---|---|---|
| Optimizer | Adam | Adam |
| Learning rate | 0.00013 | 0.00022 |
| LR Scheduler | Adaptive KL | Constant |
| $\gamma$ (discount factor) | 0.9716 | 0.9756 |
| Recurrence rollout | 8 | - |
| Clip ratio | 0.2 | 0.2 |
| Batch size | 16384 | 16384 |
| Optimization epochs | 1 | 1 |
| Entropy coefficient | 0.0156 | 0.023 |
| Value loss coefficient | 0.5 | 0.5 |
| $GAE_\lambda$ | 0.95 | 0.95 |
| MLP hidden size | 16 | 512 |
| ResNet residual blocks | 1 | 8 |
| ResNet filters | 8 | 64 |
| Attention hidden size | 16 | 512 |
| Attention heads | 4 | 8 |
| GRU hidden size | 16 | - |
| Activation function | ReLU | ReLU |
| Network Initialization | orthogonal | orthogonal |
| Rollout workers | 4 | 8 |
| Envs per worker | 4 | 4 |
| Training steps | $2 \times 10^7$ | $10^9$ |
| Episode length | 512 | 512 |
| Observation patch | $5 \times 5$ | $11 \times 11$ |
| Number of agents | 2 | 64 |

### A.2 EVALUATION SCORES

We provide the evaluation results for Dense, Directional, and Directional Negative reward functions tested for Bottleneck MAPF task. The Figures 7, 8, 9 show that SRMT has superior or comparable performance compared to the baselines.

Figure 10 shows the evaluations of the methods from the POGEMA benchmark compared to SRMT when evaluated on MovingAI maps with different numbers of agents equal to or greater than the ones used for training. SRMT was trained with 64 agents, SRMT_64_128 with a mixture of 64 and 128 agents. The results show that both SRMT models consistently outperform cooperative baselines (MAMBA and QPLEX).

Table 2: Tested reward functions. We list the reward values for achieving the goal, moving on the path toward the goal, or taking other actions (moving not in the direction of the goal and staying in the same position).

| Type | On goal | Move towards goal | Else |
|---|---|---|---|
| Directional | +1 | +0.005 | 0 |
| Sparse | +1 | 0 | 0 |
| Dense | +1 | -0.01 | -0.01 |
| Directional Negative | +1 | -0.005 | -0.01 |
| Moving Negative | +1 | -0.01 | -0.01 for moving, -0.005 for holding |

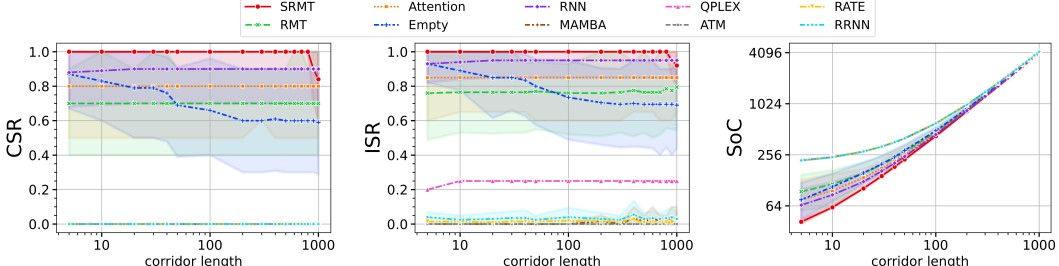

Figure 7: Trained with **Dense** reward, all models except empty core policy scale with enlarging corridor length. SRMT consistently outperforms baselines both in success rates and in the time needed to solve the task. The shaded area indicates 95% confidence intervals.

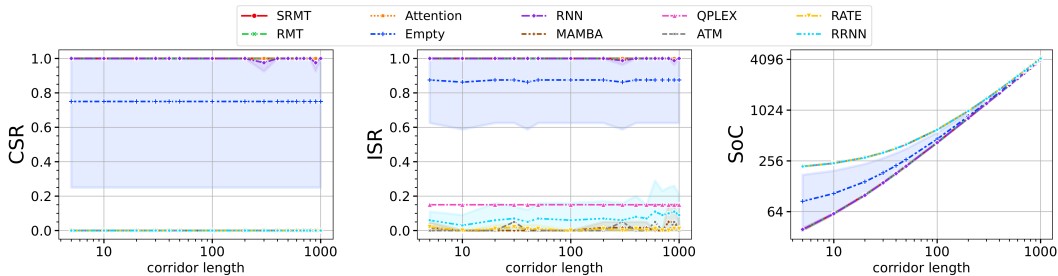

Figure 8: **Directional** reward training leads to all the methods preserving the scores for all tested corridor lengths. The shaded area indicates 95% confidence intervals.

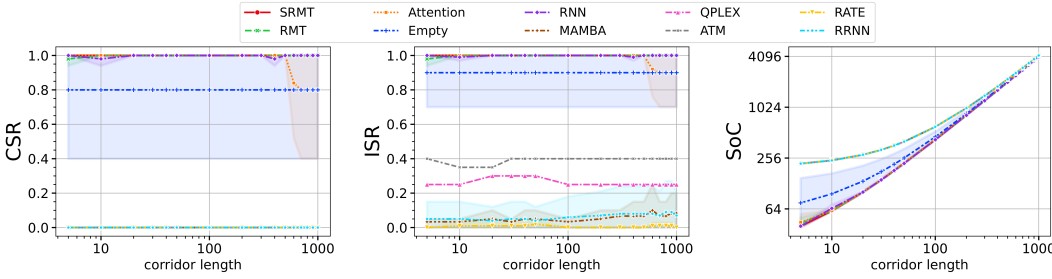

Figure 9: Results of training with **Directional Negative** reward. Vanilla attention fails to scale at corridor lengths of more than 400, compared to the SRMT which preserves the highest scores. That proves the sufficiency of the proposed SRMT architecture. The shaded area indicates 95% confidence intervals.

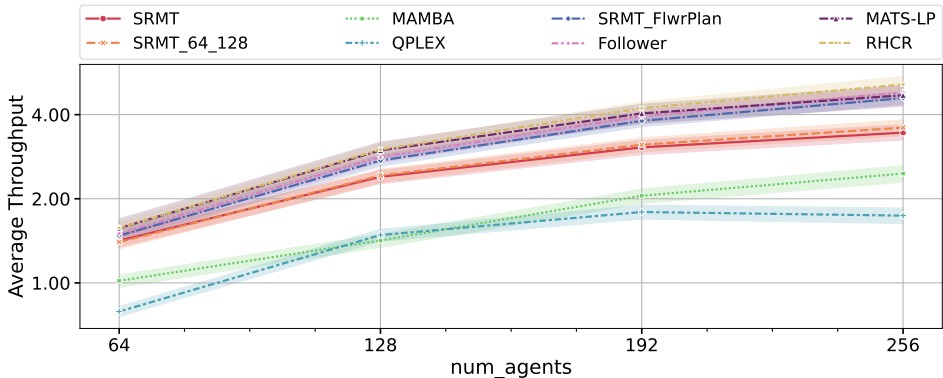

Figure 10: The evaluation of scalability of SRMT on MovingAI maps from POGEMA benchmark. The shaded area indicates 95% confidence intervals.

### A.3 MEMORY ANALYSIS

We also explored the relations between the SRMT agents' memory representations and the spatial distances between agents on the map. Fig. 11 shows that SRMT distances between memory representations are aligned with distances between agents for different corridor lengths. Starting the episode, the agents move closer to each other quickly, and the respective cosine distances decrease significantly. Then, agents face each other in the environment (marked with a triangle on Fig. 11)

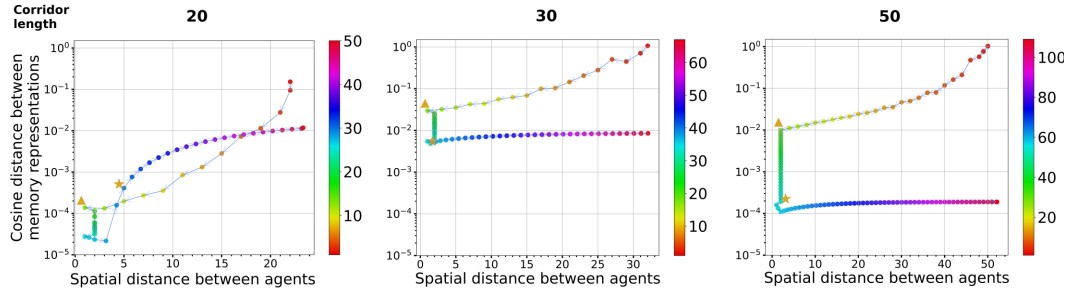

Figure 11: **SRMT memory distances are aligned with the distances between agents.** The figure shows how cosine distances between agents' memory vectors are related to the Euclidean distances between agents on the map for SRMT. The triangle marks the step when agents face each other in the environment, the star shows the episode step when the first goal was achieved. The color bar shows the step number.

and move in the same direction, keeping the spatial distance between them constant. Next, after the moment when one of the agents reaches its goal and disappears from the environment (marked with a star), the other agent moves away to reach the goal. This part of the episode is depicted as increasing memory distance at the end of the episode.