# OpenReview forum: "Shared Memory for Multi-agent Lifelong Pathfinding"
_ICLR.cc/2025/Conference — Submitted to ICLR 2025_

### Official Review · Reviewer_r4zr · 2024-10-28

**Soundness:** 2
**Presentation:** 2
**Contribution:** 2
**Rating:** 5
**Confidence:** 3

**Summary:**

The paper proposes a global shared recurrent memory transformer (SRMT) mechanism for multiagent reinforcement learning to address the multiagent pathing finding problem. Specifically, SRMT uses self-attention to aggregate agent memory and observation history while utilizing cross-attention to aggregate the shared memory from other agents to help coordination. Results on a toy bottleneck navigation task and a set of maze environments from the POGEMA benchmark show that SRMT outperforms various baselines.

**Strengths:**

1.	The motivation for using a global shared memory to help coordination and the idea of using the transformer to implement it are clear.
2.	The background is clearly explained and the related works are well discussed.

**Weaknesses:**

1.	It seems that a lot baselines are missing. For example, in the Bottleneck Task, only some basic memory mechanisms from single-agent RL are compared while more advanced memory mechanisms such as relational memory [1] and AMRL [2] from the single-agent RL domain are not compared.
2.	At the same time, although some works about MARL memory such as RATE and ATM are discussed in Section 2.2, they are not compared in the experiments.
3.	The ablation study to validate each component of the proposed SRMT is not given.
4.	There are some typos. In Line 36, “MAPF” is not defined.

References

[1] Adam Santoro, Ryan Faulkner, David Raposo, Jack Rae, Mike Chrzanowski, Théophane Weber, Daan Wierstra, Oriol Vinyals, Razvan Pascanu, and Timothy Lillicrap. Relational Recurrent Neural Networks. In Proceedings of the 32nd International Conference on Neural Information Processing Systems, 2018.

[2] Jacob Beck, Kamil Ciosek, Sam Devlin, Sebastian Tschiatschek, Cheng Zhang, and Katja Hofmann. Amrl: Aggregated memory for reinforcement learning. In International Conference on Learning Representations, 2020.

**Questions:**

1.	Could the authors give the number of network parameters of each method? As SRMT uses transformers and ResNet, it may obtain advantages by more network parameters.
2.	Could SRMT scale well with the number of agents? If the number of agents increases, will the training time become much longer?
3.	Why does MAMBA with discrete communication protocol outperform SRMT in some scenarios? Does it mean that the global shared memory is not always the best choice? If yes, how could we choose the right method for the multiagent path-finding problem?

---

> ### Author Response · Authors · 2024-11-25
> **Response to Reviewer r4zr, part 1.**
>
> Dear Reviewer r4zr,
>
> We sincerely appreciate your time and constructive comments.
>
> In the following, we would like to address your concerns separately.
>
> > **W1: It seems that a lot baselines are missing. For example, in the Bottleneck Task, only some basic memory mechanisms from single-agent RL are compared while more advanced memory mechanisms such as relational memory [1] and AMRL [2] from the single-agent RL domain are not compared.**
>
> > **W2: At the same time, although some works about MARL memory such as RATE and ATM are discussed in Section 2.2, they are not compared in the experiments.**
>
> Thank you for your suggestions.
>
> We have reproduced the relational recurrent neural network (RRNN), RATE, and ATM approaches and evaluated them on the Bottleneck task. The results are presented in Figures 3, 4, 7, 8, and 9 of the updated submission. As shown in Figure 3, SRMT outperformed all the implemented baseline approaches trained under different reward schemes.
>
> During the implementation of RRNN, RATE, and ATM, we observed a shared behavior among these architectures: their memory state is initialized using pre-defined values (e.g., a unit vector or values sampled from a standard normal distribution). In contrast, SRMT initializes its memory state with values derived from the first step of the episode, based on the initial observations.
>
> To better understand the significant difference in CSR scores between baseline memory approaches and SRMT, we conducted an additional experiment. Specifically, we modified the initialization of the RATE memory state to use values generated from the agent’s initial observation, similar to SRMT. The results, depicted in Figure 4 (RATE_gen), show a notable performance improvement under the Moving Negative reward scheme compared to the original RATE implementation.
>
> Training memory-based baselines (RRNN, RATE, ATM) on the POGEMA benchmark task requires significantly more time than on the Bottleneck task and will not be completed within the rebuttal period. However, we are actively running these experiments and will include the results in a future update as soon as they are ready.
>
> > **W3: The ablation study to validate each component of the proposed SRMT is not given.**
>
> Thank you for pointing this out.
>
> The Bottleneck task results not only demonstrate the effectiveness of SRMT in two-agent coordination but also serve as an ablation study to isolate and highlight the role of shared memory within SRMT. Specifically:
>
> - **RMT** uses the same mechanisms as SRMT to generate and process individual memory states for agents but does not allow sharing of memory states between agents.
> - **Attention Core** removes both individual and shared memory, testing an architecture without memory components.
> - In the **Empty Core** setup, the core network is completely removed from the policy model, creating a direct connection between the spatial encoder and the actor-critic decoder.
>
> The evaluation results in Figure 3 show that SRMT achieves the highest scores across all setups, particularly in scenarios with sparse rewards, where the task is more challenging for agents to learn. Additionally, models with shared memory demonstrate greater stability across runs, as evidenced by tighter confidence intervals compared to methods without shared memory.
>
> We have clarified the ablation study and included new memory baselines in the updated version of the manuscript to strengthen the analysis.
>
> > **W4: There are some typos. In Line 36, “MAPF” is not defined.**
>
> Thank you for your notice. We carefully revised our manuscript and fixed typos.
>
>
> [1] Santoro et al. Relational Recurrent Neural Networks. In Proceedings of the 32nd International Conference on Neural Information Processing Systems, 2018.
>
> [2] Beck et al. Amrl: Aggregated memory for reinforcement learning. In International Conference on Learning Representations, 2020.

---

> ### Author Response · Authors · 2024-11-25
> **Response to Reviewer r4zr, part 2.**
>
> > **Q1: Could the authors give the number of network parameters of each method? As SRMT uses transformers and ResNet, it may obtain advantages by more network parameters.**
>
> Number of trainable parameters:
> | Model          | Bottleneck task |    POGEMA benchmark    |
> |----------------|:---------------:|:----------------------:|
> | SRMT           |       271k      |           17M          |
> | RMT            |       271k      |            -           |
> | Attention Core |       271k      |            -           |
> | Empty Core     |        5k       |            -           |
> | RNN Core       |        6k       |            -           |
> | MAMBA          |        6M       |           6M           |
> | QPLEX          |       318k      |          318k          |
> | ATM            |       349k      |            -           |
> | RATE           |       272k      |            -           |
> | RRNN           |        8k       |            -           |
> | Follower       |        -        |           5M           |
> | MATS-LP        |        -        |          161k          |
> | RHCR           |        -        | no training parameters |
>
> The evaluation results on the Bottleneck task show that SRMT outperforms the models of comparable size (RATE and QPLEX) and the models with a larger number of trainable parameters (MAMBA and ATM).
>
> The POGEMA evaluations demonstrate that the size of the trainable model has little effect on the final performance metrics. For example, the best Performance value is obtained by the RHCR algorithm which does not require training. Methods that use centralized path planning strategies (Follower and MATS-LP) have higher Performance scores than SRMT while having a smaller number of parameters.
>
> > **Q2: Could SRMT scale well with the number of agents? If the number of agents increases, will the training time become much longer?**
>
> For the POGEMA benchmark task, we trained two SRMT models with 64 agents and with a mixture of 64 and 128 agents on Mazes type of maps. Following the benchmark evaluation procedure, for MovingAI evaluations, all models were tested with greater numbers of agents (64, 128, 192, 256) on each map compared to the training setting. We added the detailed performance for each method for each number of agents in Appendix A.2 Figure 10 of the revised manuscript. The results show that SRMT consistently outperforms communicative baselines (MAMBA and QPLEX) when evaluated with greater agent populations.
>
> To further address the scalability assessment, the POGEMA benchmark includes specific evaluations on the Scalability metric, designed to show how the runtime of the method changes with the growing number of agents. Figure 6 shows that SRMT has better scalability than MATS-LP which does not use communication and search-based planner RHCR. SRMT demonstrates comparable performance to methods with centralized training (MAMBA, QPLEX) and hybrid Follower with decentralized training and centralized path-planning.
>
> Considering training models for the same number of environment steps, training time does not depend on the number of agents. Training with different numbers of agents will result in differences in effective batch sizes for policy network training.
>
> For training SRMT we use the Sample Factory codebase that provides effective implementations of the environment simulation and the collection of trajectories for policy network training.

---

> ### Author Response · Authors · 2024-11-25
> **Response to Reviewer r4zr, part 3.**
>
> > **Q3.1: Why does MAMBA with discrete communication protocol outperform SRMT in some scenarios?**
>
> Thank you for your thoughtful questions.
> Considering the evaluation on the Warehouse map, SRMT trained with 64 agents achieves an Average Throughput of $1.38\pm0.02$, SRMT trained on a mixture of 64 and 128 agents scores $1.43\pm 0.02$, and MAMBA achieves $1.50\pm 0.03$ Average Throughput. It is worth noting that the Warehouse evaluation was conducted on a single obstacle configuration, with different random seeds for each number of agents. In contrast, the rest of the evaluation tasks (Mazes, MovingAI, Puzzles, Random) use between 8 and 128 different obstacle configurations for evaluation.
>
> As a result, the Warehouse task has significantly lower variety in its evaluation data. Compared to tasks with evaluation maps of similar sizes (e.g., the Warehouse map is $33\times 46$, while Random and Maze maps range from $17\times 17$ to $21\times 21$, and MovingAI maps are $64\times 64$), the Warehouse results exhibit the tightest error bars. This reduced variance may arise from the single-configuration evaluation setup and could contribute to the observed difference in SRMT and MAMBA scores.
>
> The POGEMA benchmark includes the Warehouse map primarily to calculate the Scalability metric, which assesses how the runtime of different methods scales with a larger number of agents. Thus, while MAMBA marginally outperforms SRMT in this scenario, it is important to consider the broader evaluation results across diverse tasks.
>
> > **Q3.2: Does it mean that the global shared memory is not always the best choice? If yes, how could we choose the right method for the multiagent path-finding problem?**
>
> To answer your question, we refer to the comprehensive evaluations of the Bottleneck task, showing the superior performance and scalability of SRMT compared to other methods. Also, POGEMA evaluations show that SRMT improves over the communication baselines (MAMBA and QPLEX) on the maps of size comparable to the training one and significantly bigger maps (SRMT was trained on maps of size $65\times 65$, and evaluated on maps of size ranging from $17\times 17$ on Random and Mazes to $256\times 256$ on MovingAI). We also showed how SRMT performance is consistently superior to the communicative baselines with greater numbers of agents in Appendix A.2 Figure 10 of the revised manuscript.
>
> MAMBA’s structured, discrete communication channels allow agents to exchange specific, targeted information, e.g., intended movements or status updates, which can be particularly effective in densely populated environments. In contrast, SRMT relies on a general-purpose shared recurrent memory that agents learn to use adaptively.
>
> We agree that it is a very important question. As centralization is not always feasible, therefore decentralized methods such as SRMT are more flexible and provide a strong alternative.
>
> -----------------
> We appreciate your valuable feedback and have updated our manuscript according to your suggestions:
> - added the Relational RNN, RATE, and ATM baselines;
> - added the results of an additional evaluation for RATE with SRMT-like initialization of agent memory state that significantly improved RATE performance;
> - clarified that RMT, Attention Core, and Empty Core serve as the ablations of SRMT;
> - revised the text and fixed typos;
> - added the Figure 10 in Appendix A.2 that illustrates the scalability of SRMT with the number of agents.
>
> We hope these revisions will clarify our contributions and positively influence your assessment of our work.

---

> ### Comment · Reviewer_r4zr · 2024-11-26
>
> The reviewer appreciates the authors' feedback. Most of my concerns are addressed, but I think this paper's novelty struggles to meet this conference's criteria and it also needs polishing to make the details easy to understand.

---

> > ### Author Response · Authors · 2024-11-26
> >
> > Thank you for your feedback and for acknowledging that we addressed most of your concerns. We would like to emphasize that we were unable to find any prior studies in the literature that utilize global shared memory for multi-agent reinforcement learning (MARL) and planning, which leads us to believe that our contribution is indeed novel. If you are aware of publications exploring this idea, we would greatly appreciate it if you could share them, so we can compare and address your concern regarding the lack of novelty.

---

> > > ### Comment · Reviewer_r4zr · 2024-11-26
> > >
> > > The idea of using global shared memory is straightforward in MARL such as the global hidden state in [1]. At the same time, this raises a question about the communications in the setting of decentralized execution, and authors should also discuss and compare with the MARL communication works.
> > >
> > > References
> > >
> > > [1] Hu, S., Zhu, F., Chang, X., & Liang, X. (2021). UPDeT: Universal Multi-agent RL via Policy Decoupling with Transformers. International Conference on Learning Representations. https://openreview.net/forum?id=v9c7hr9ADKx

---

> > > > ### Author Response · Authors · 2024-11-27
> > > >
> > > > Thank you for providing this reference, it helped us to understand better the novelty of our contribution. Differences between SRMT and UPDeT [1] can be summarized as follows:
> > > > - In SRMT shared memory enables agents to access the information about the transitions of agents that are both inside and outside the agent’s view range. In UPDeT, the agent has information about fellow agents located within the agent’s view range. This difference highlights the ability of SRMT to provide a more global perspective for agents' decision-making process.
> > > > - In UPDeT, the global hidden state consists of hidden vectors, each of which tracks the history of observations of a single agent similar to RMT, ATM, RATE, RRNN, and other single-agent RL memory architectures. Term ‘global’ means a hidden vector stores all the information available to a single agent within its view range. Naming such a hidden state ‘global’ might sound slightly misguiding when compared with other MARL communication-related works. In contrast, the SRMT shared memory state contains memory vectors for all the agents in the multi-agent system and is fully available to each agent.
> > > > - UPDeT and related works (MAT [2], ACUTE [3], TransMix [4], UNSR [5]) have not been applied to MAPF and have not been compared with models developed specifically for MAPF (such as Follower, MATS-LP, RHCR) as opposed to SRMT.
> > > > - The hidden state in UPDeT is designed to hold the information of the action-observation history, while the SRMT’s shared memory is used as a channel for inter-agent networking, serving the different goal.
> > > >
> > > > We also acknowledge the valid questions that have been raised about communication work in MARL. Our results include comparisons with centralized training with decentralized execution (CTDE) MARL methods that incorporate communication, such as QPLEX and MAMBA. SRMT outperformed these methods in all environments in the Bottleneck task and POGEMA benchmark and on all metrics except congestion management, where SRMT still showed competitive performance.
> > > >
> > > > A key distinction of SRMT is that it uses a general-purpose shared recurrent memory and relies only on local agents' observations. Communication may be considered as one of the possible uses of this shared memory, but it is not explicitly predefined or structured. This flexibility distinguishes SRMT from methods that rely on fixed communication protocols and allows it to dynamically adapt to different scenarios.
> > > >
> > > > We will add the discussion of novelty into the final version of the manuscript.

---

> > > > > ### Author Response · Authors · 2024-12-02
> > > > > **Request to revisit responses**
> > > > >
> > > > > We kindly request that you revisit our detailed responses, where we have made every effort to address your concerns comprehensively. Your feedback has been invaluable in guiding this process, and we hope our responses reflect our commitment to engaging thoughtfully with the review process and enhancing the quality of our paper.
> > > > >
> > > > > Thank you once again for your insightful comments, and we sincerely hope the updates meet your expectations.

---

### Official Review · Reviewer_fa5a · 2024-11-04

**Soundness:** 3
**Presentation:** 3
**Contribution:** 3
**Rating:** 8
**Confidence:** 4

**Summary:**

This paper introduces the Shared Recurrent Memory Transformer (SRMT), a novel model in multi-agent reinforcement learning designed for multi-agent lifelong pathfinding tasks. SRMT extends memory transformers to decentralized multi-agent environments by pooling individual agent memories into a shared memory space, allowing agents to indirectly share information and coordinate. The model is tested in various pathfinding tasks, including bottleneck navigation and complex environments from the POGEMA benchmark. SRMT demonstrates superior performance in coordination and generalization, particularly in high-density and partially observable environments.

**Strengths:**

1. The SRMT model is an adaptation of memory transformers to multi-agent settings, facilitating indirect communication among agents through a shared memory. This approach addresses a significant challenge in decentralized coordination by leveraging shared recurrent memory, which is unique compared to conventional communication strategies.
2. The paper provides a rigorous evaluation of SRMT on multiple benchmark tasks, including POGEMA and bottleneck navigation. The use of diverse reward settings (e.g., sparse, directional) further strengthens the experimental framework, revealing SRMT’s adaptability in various coordination scenarios.
3. The architecture and methods are clearly explained, supported by diagrams and flowcharts that help clarify SRMT’s working mechanism. The comparisons with baselines and the explanation of the multi-agent Markov decision process formulation are presented in a straightforward and understandable manner.
4. SRMT’s ability to handle decentralized pathfinding without explicit communication protocols has considerable implications for real-world applications, particularly in settings where communication might be unreliable or costly. Its effectiveness across different maps and scenarios demonstrates potential for scalability in complex, large-scale environments.

**Weaknesses:**

1. While SRMT performs well on small to medium-sized environments, its scalability to very large maps or highly dense environments remains uncertain. The evaluation could be extended to more challenging settings, particularly with greater agent populations or larger obstacles, to fully assess SRMT’s scalability.
2. While SRMT is designed for decentralized systems, it would be beneficial to see comparisons with centralized approaches on key metrics to understand the trade-offs better, particularly in environments that demand high coordination.
3. While the paper claims that shared memory improves coordination, additional analysis on how shared memory influences individual agent behavior would provide a deeper understanding. An ablation study removing the shared memory aspect could further validate its impact on SRMT’s performance.
4. The model's performance varied across different reward structures, and while this is discussed, a more detailed exploration of how reward shaping influences learning would strengthen the analysis. This would help in tailoring SRMT to tasks where only sparse rewards are available.

Missing references (MARL with local information). I believe these are quite recent papers and work in a similar setting as mentioned in the related works section.

[1]: Hu, Y., Fu, J., & Wen, G. (2023). Graph soft actor–critic reinforcement learning for large-scale distributed multirobot coordination. *IEEE transactions on neural networks and learning systems*.

[2]: Nayak, S., Choi, K., Ding, W., Dolan, S., Gopalakrishnan, K., & Balakrishnan, H. (2023, July). Scalable multi-agent reinforcement learning through intelligent information aggregation. In *International Conference on Machine Learning* (pp. 25817-25833). PMLR.

**Questions:**

1. How well does SRMT scale with an increased number of agents or more complex map structures? Additional experiments in larger environments could help evaluate its robustness in real-world applications.
2. Would SRMT benefit from combining shared memory with limited explicit communication for certain high-density environments?
3. How does shared memory impact the decision-making process for individual agents? Further analysis on memory usage patterns and shared memory dynamics could provide insights into SRMT’s internal coordination mechanisms.
4. Does SRMT allow for integration with hierarchical pathfinding methods, such as combining local and global pathfinding strategies?

---

> ### Author Response · Authors · 2024-11-25
> **Response to Reviewer fa5a, part 1.**
>
> Dear Reviewer fa5a,
>
> We sincerely appreciate your time and constructive comments.
> Thank you for recognizing the novelty of our approach and the potential of SRMT in complex real-world applications.
>
> In the following, we would like to address your comments and suggestions separately.
> > **W1: While SRMT performs well on small to medium-sized environments, its scalability to very large maps or highly dense environments remains uncertain. The evaluation could be extended to more challenging settings, particularly with greater agent populations or larger obstacles, to fully assess SRMT’s scalability.**
>
> > **Q1: How well does SRMT scale with an increased number of agents or more complex map structures? Additional experiments in larger environments could help evaluate its robustness in real-world applications.**
>
> For the POGEMA benchmark task, we trained two SRMT models with 64 agents and with a mixture of 64 and 128 agents on Mazes maps of size 65x65. Following the benchmark evaluation procedure, for MovingAI evaluations we used 128 maps of size 256x256 with different configurations of obstacles.
> Moreover, all models were tested with greater numbers of agents (64, 128, 192, 256) on each map compared to the training setting. This evaluation estimates how models perform for higher agent densities because map size is fixed. We added the detailed performance for each method for each number of agents in Appendix A.2 Figure 10 of the revised manuscript. The results show that SRMT consistently outperforms communicative baselines (MAMBA and QPLEX) when evaluated with higher agent densities.
>
> To further address the scalability assessment, the POGEMA benchmark includes specific evaluations on the _Scalability_ metric, designed to show how the runtime of the method changes with the growing number of agents. Figure 6 shows that SRMT has better scalability than MATS-LP which does not use communication and search-based planner RHCR. SRMT demonstrates comparable performance to methods with centralized training (MAMBA, QPLEX) and hybrid Follower with decentralized training and centralized path-planning.
>
> We appreciate your suggestion to explore even more challenging settings. If you have specific environment types in mind that would further test SRMT, we would be grateful to incorporate these environments in future work.
> > **W2: While SRMT is designed for decentralized systems, it would be beneficial to see comparisons with centralized approaches on key metrics to understand the trade-offs better, particularly in environments that demand high coordination.**
>
> Thank you for highlighting the importance of comparing with centralized methods. We use RHCR as a centralized search-based baseline and evaluate it on the POGEMA benchmark to compare SRMT performance relative to a centralized approach (see Figures 5, 6).
>
> RHCR scores close to 100% in Performance and Pathfinding metrics, and is the best in Cooperation and Out-of-Distribution metrics. However, centralization has notable trade-offs: RHCR scores poorly on Congestion and Scalability metrics, performing among the worst in these areas compared to other methods. In contrast, SRMT can handle high-density environments (as evidenced by the Congestion scores) more effectively than centralized RHCR.
>
> We appreciate your feedback, as it has helped us clarify the trade-offs between the centralized RHCR and decentralized SRMT in our results.

---

> ### Author Response · Authors · 2024-11-25
> **Response to Reviewer fa5a, part 2.**
>
> > **W3: While the paper claims that shared memory improves coordination, additional analysis on how shared memory influences individual agent behavior would provide a deeper understanding. An ablation study removing the shared memory aspect could further validate its impact on SRMT’s performance.**
>
> > **Q3: How does shared memory impact the decision-making process for individual agents? Further analysis on memory usage patterns and shared memory dynamics could provide insights into SRMT’s internal coordination mechanisms.**
>
> Thank you for your insightful feedback.
> The bottleneck task results provide valuable evidence of SRMT's effectiveness in two-agent coordination and can also be viewed as an ablation study highlighting the role of the proposed shared memory mechanism. Specifically:
> - **RMT:** Agents use individual memory representations locally without sharing them.
> - **Attention Core:** Memory is entirely removed from the core part of the policy model, testing a memoryless architecture.
> - **Empty Core:** The core network is completely removed from the policy model.
>
> The results of these ablation studies, shown in Figure 3, demonstrate that shared memory is a crucial component of SRMT, particularly in sparse reward scenarios where tasks are more challenging for agents to learn. These findings underscore the positive impact of the shared recurrent memory mechanism on coordination performance.
>
> Additionally, SRMT shows greater stability across runs, as evidenced by the tighter confidence intervals compared to methods without shared memory.
>
> In the section Memory Analysis of Appendix we provide insights of inner workings of shared memory during task execution demonstrating that distances between memory representations are aligned with distances between agents and modes of interaction. Starting the episode, the agents move closer to each other quickly, and the respective cosine distances between memory representations decrease significantly. This decrease continues as agents face each other in the environment and move together in the same direction along the corridor. Next, after the moment when one of the agents reaches its goal and disappears from the environment, the memory representations slightly diverge as the remaining agent moves away to reach the goal. We appreciate your suggestions and hope this explanation addresses your concerns. Let us know if further clarification is needed.
>
> > **W4: The model's performance varied across different reward structures, and while this is discussed, a more detailed exploration of how reward shaping influences learning would strengthen the analysis. This would help in tailoring SRMT to tasks where only sparse rewards are available.**
>
> Thank you for your valuable feedback.
>
> The motivation behind our experiments with different reward structures was to evaluate how effectively agents can leverage shared memory to solve specific navigational sub-tasks induced by these reward schemes. Below, we provide additional discussion on how reward shaping influences learning and the role of shared memory in these scenarios:
> - Sparse Reward provides no explicit constraints on the agent's movement to achieve its goal. Agents must independently discover effective strategies, making shared memory crucial for avoiding potential collisions and improving coordination.
> - Dense and Moving Negative Rewards generally discourage unnecessary movements, with Moving Negative specifically incentivizing agents to minimize transitions and, in some cases, freeze in place. Shared memory is beneficial in these scenarios as it enables agents to coordinate their movements more efficiently, avoiding redundant transitions and optimizing performance under transition penalties.
> - Directional and Directional Negative Rewards encourage agents to move directly toward the goal, increasing the likelihood of collisions in narrow corridors. Here, shared memory plays a critical role by providing information about the movement history of other agents, allowing for better collision avoidance and more efficient decision-making.
>
> We will expand the discussion in the paper to include these insights and further elaborate on the influence of reward shaping on learning and coordination.

---

> ### Author Response · Authors · 2024-11-25
> **Response to Reviewer fa5a, part 3.**
>
> > **Q2: Would SRMT benefit from combining shared memory with limited explicit communication for certain high-density environments?**
>
> We appreciate your proposition. Indeed, the combination of limited explicit communication and shared memory could help agents exchange information more directly. However, explicit communication provides additional costs for the practical implementation of MARL and makes agents less independent in their decision-making process, which reduces the decentralization of the multi-agent system. This is a promising direction for future work.
>
> >**Q4: Does SRMT allow for integration with hierarchical pathfinding methods, such as combining local and global pathfinding strategies?**
>
> Yes, SRMT memory is implemented as an integral part of the agent's policy network, allowing it to seamlessly integrate with various pathfinding strategies, including hierarchical approaches that combine local and global pathfinding. This flexibility allows SRMT to work with pathfinding strategies embedded in the agent's interactions with the environment, supporting both local decision-making and broader navigation goals.
>
> Thank you for this great suggestion - it opens a promising direction for further improving SRMT's coordination capabilities in complex environments.
>
> > **Missing references (MARL with local information). I believe these are quite recent papers and work in a similar setting as mentioned in the related works section.**
>
> We appreciate your suggestions. We added the proposed references to a related works section of the manuscript.
>
> ---------------
>
> We appreciate your valuable feedback and have updated our manuscript according to your suggestions:
> - added the detailed illustration in Appendix A.2 Figure 10 of the manuscript reflecting the SRMT scalability with the number of agents;
> - clarified which models were used as ablations to SRMT.
> - added the proposed references to the related works section of the manuscript.

---

> > ### Comment · Reviewer_fa5a · 2024-11-27
> >
> > Thanks a lot for addressing my reviews and for improving the paper's quality. I am not sure if I see both the references I pointed out in my review in the related works sections. I believe there are a few other papers as well that are quite similar to these and have operated in similar paradigms.

---

> ### Author Response · Authors · 2024-11-27
>
> Thank you for your answer.
>
> We re-loaded the rebuttal version of the manuscript.
> Both suggested references are mentioned in the end of the first paragraph of section 2.1.
>
> We would appreciate it if you let us know if there are other references you can recommend to add to the related works section.

---

> > ### Comment · Reviewer_fa5a · 2024-11-28
> >
> > I really appreciate the thorough response by the authors. I believe this paper is a good contribution to the MARL community.
> >
> > Another reference that I think is relevant:
> > Agarwal, A., Kumar, S., and Sycara, K. P. Learning transferable cooperative behavior in multi-agent teams. CoRR, abs/1906.01202, 2019. URL http://arxiv.org/abs/1906.01202

---

> > > ### Author Response · Authors · 2024-11-29
> > >
> > > We are deeply grateful for your insightful review. Your high scores and constructive feedback are incredibly encouraging and demonstrate a genuine engagement with our research.
> > >
> > > We updated the manuscript to include the proposed reference.

---

### Official Review · Reviewer_766H · 2024-11-04

**Soundness:** 2
**Presentation:** 3
**Contribution:** 2
**Rating:** 3
**Confidence:** 3

**Summary:**

This work considers the application of a shared memory mechanism to the MAPF setting.

**Strengths:**

- The writing is generally clear and polished.
- The approach is well-grounded in prior literature, and the algorithmic details are well-explained.
- Figure 1 is a useful complement to the written algorithmic details, and makes it easy to understand the method at a glance.
- Figure 10 analysis is nice.

**Weaknesses:**

* It is hard to get a relative sense of the competitiveness of this approach. The baselines did not feel particularly well-motivated, and MARL communication works, which I'd argue share a similar goal, were not used as baselines (e.g. \[1\])
* More generally, I am left not knowing exactly what I should take away from the results—Figure 5 seems to show that SRMT and variants achieve modest results compared to baselines (and the baselines used are not motivated or described in sufficient detail).
* \[2\] I consider this a necessary work to acknowledge, given it is one of the first works discussing the use of attention in MARL
* Nitpicks:
	* I cannot interpret the error bars in Figure 4—it is too muddled.
	* Despite the writing overall being clear, the language could be tightened somewhat; e.g. L043: "has to reach its goal" is quite colloquial; also contraction in L497. I recommend combing through the paper and essentially asking each word/phrase to justify itself—and to be as specific as possible, avoiding colloquialisms.

\[1\] Jakob Foerster, Ioannis Alexandros Assael, Nando de Freitas, and Shimon Whiteson. Learning to communicate with deep multi-agent reinforcement learning. In D. Lee, M. Sugiyama, U. Luxburg, I. Guyon, and R. Garnett (eds.), *Advances in Neural Information Processing Systems, volume 29*. Curran Associates, Inc., 2016. URL https://proceedings.neurips.cc/paper_ files/paper/2016/file/c7635bfd99248a2cdef8249ef7bfbef4-Paper.pdf.

\[2\] Iqbal, S. &amp; Sha, F.. (2019). Actor-Attention-Critic for Multi-Agent Reinforcement Learning. <i>Proceedings of the 36th International Conference on Machine Learning</i>, in <i>Proceedings of Machine Learning Research</i> 97:2961-2970 Available from https://proceedings.mlr.press/v97/iqbal19a.html.

**Questions:**

- Following up on a weakness above: Why was this approach not evaluated against any MARL baselines that implement communication channels between agents?

---

> ### Comment · Reviewer_766H · 2024-11-24
>
> Contextualizing one's work relative to prior work is one of the most important aspects of an academic paper. I would like to emphasize that many baselines and related works seem to be missing (a concern echoed by the other reviewers). Given the lack of engagement from the authors to address this concern, I am lowering my score to signal the unreadiness of this paper for publication at ICLR.
>
> Otherwise, the work is generally compelling, and I hope the authors will take reviewer feedback seriously and resubmit in a later conference.

---

> ### Author Response · Authors · 2024-11-25
>
> Dear Reviewer 766H,
>
> We highly appreciate your time and effort to review our submission.
>
> Your comments and suggestions are valuable to us.
>
> We wanted to acknowledge that we are actively preparing a response to address the points you have raised. Some of the experiments requested by the reviewers are currently running, and we are waiting for them to be completed to provide you with a comprehensive response as soon as possible.
>
> We greatly value the opportunity to engage in this scientific dialogue and look forward to addressing your concerns in detail through the formal author response.

---

> ### Author Response · Authors · 2024-11-25
> **Response to Reviewer 766H, part 1.**
>
> Dear Reviewer 766H,
>
> We sincerely appreciate your time and constructive comments.
>
> Thank you for recognizing the grounding of our method in prior literature, the clarity in the proposed method explanation and paper writing, and the provided memory analysis.
>
> In the following, we would like to address your concerns separately.
> > **W1: It is hard to get a relative sense of the competitiveness of this approach. The baselines did not feel particularly well-motivated, and MARL communication works, which I'd argue share a similar goal, were not used as baselines (e.g. [1])**
>
> With our baselines, we aimed to cover a wide range of coordination-related approaches commonly used for MAPF in the literature: fully decentralized methods such as MAMBA, Follower, and MATS-LP; fully centralized RHCR; QPLEX that allows centralized training with decentralized execution. Also, MAMBA represents the approach with communication, and Follower and MATS-LP use centralized path-planning strategies.
>
> Baselines such as RMT, Attention core, and Empty core serve as ablations of SRMT architecture. In RMT agents use individual memory representations without sharing them. In the Attention core method, the memory is completely removed from the core part of the policy model to test the memoryless architecture. Finally, in the Empty core, the core network is removed from the policy. The results of these ablation methods are in Figure 3 and show that shared memory is a key component of SRMT, especially in the sparse rewarding scenario, where the task is harder for agents to learn.
>
> Considering MAMBA and QPLEX as methods that allow communication, we added their evaluations on the Bottleneck task into the updated version of the paper (Figures 3,4,7,8,9). The results show that SRMT consistently outperforms both methods on the Bottleneck task. On the POGEMA benchmark, SRMT outperforms QPLEX on all maps and outperforms MAMBA on all maps except Warehouse, where the resulting scores are comparable.
>
> We also added MARL approaches that employ memory mechanisms (RATE [2], RRNN [3], ATM [4]) as baselines. Considering the limited time of the rebuttal period, we trained and evaluated them on the Bottleneck task only. Training these methods on the POGEMA benchmark requires more time and can not be completed until the end of the rebuttal period. We are running these experiments and will add the results as soon as they are ready.
>
> We considered RIAL and DIAL methods introduced in [1]. These were proposed to solve _sequential_ multi-agent decision-making problems with a discrete limited-bandwidth communication channel. Such approaches require a _single_ agent to be active at each time step. In multi-agent pathfinding tasks, all agents perform actions simultaneously at each time step, making it impossible to directly apply the communication protocol proposed in RIAL and DIAL.
> > **Q1: Following up on a weakness above: Why was this approach not evaluated against any MARL baselines that implement communication channels between agents?**
>
> Thank you for your comment and question. To address the communication baselines on bottleneck environments we added evaluations of MAMBA and QPLEX methods as mentioned in the response to W1 (see updated Figures 3, 4, 7, 8, and 9 in the manuscript). We want to highlight that the results of these methods on the POGEMA benchmark were already present in Figures 5, and 6, and SRMT shows better results on 4/5 POGEMA environments and is much better than MAMBA and QPLEX in Performance, Pathfinding, Cooperation and Out-of-distribution metrics.
>
> Indeed, we fully agree that MARL models with communication and our shared memory approach solving the same problem of coordination. However, MARL has explicit communication between agents but SRMT relies on an implicit sharing of embeddings which might be trained to contain global information about the environment state and actions of agents. Access to representations in a shared global memory allows agents to dynamically integrate them with hidden embeddings of agent’s history of local observations, rather than reading messages as a part of state observation. Here, we test a hypothesis that “soft” hidden representations in memory might support richer and more effective communication compared to explicit exchange of messages.
>
> [1] Foerster et al. Learning to communicate with deep multi-agent reinforcement learning. In D. Lee, M. Sugiyama, U. Luxburg, I. Guyon, and R. Garnett (eds.), Advances in Neural Information Processing Systems, volume 29. Curran Associates, Inc., 2016.
>
> [2] Cherepanov et al. Recurrent action transformer with memory, 2024. Arxiv:2306.09459.
>
> [3] Santoro et al. Relational Recurrent Neural Networks. In Proceedings of the 32nd International Conference on Neural Information Processing Systems, 2018.
>
> [4] Yang et al. Transformer-based working memory for multiagent reinforcement learning with action parsing. Advances in Neural Information Processing Systems, 35:34874–34886, 2022.

---

> ### Author Response · Authors · 2024-11-25
> **Response to Reviewer 766H, part 2.**
>
> > **W2: More generally, I am left not knowing exactly what I should take away from the results—Figure 5 seems to show that SRMT and variants achieve modest results compared to baselines (and the baselines used are not motivated or described in sufficient detail).**
>
> Thank you for your notice. We added the motivations and details on our baselines (RHCR, Follower, MATS-LP, MAMBA, QPLEX) to related works in Section 2.1.
> In Figure 5, MAMBA and QPLEX are cooperative MARL approaches that allow communication between agents similar to SRMT (MAMBA uses a Transformer-based communication block and QPLEX has centralized training with a decentralized execution). The evaluation results show that SRMT outperforms MAMBA and QPLEX.
> Follower, MATS-LP, and RHCR are three top-performing models from the POGEMA benchmark considered as the upper bounds for our evaluations. Follower and MATS-LP are hybrid methods that use centralized path planning during training and have decentralized execution. RHCR is a centralized search-based planner that does not require training.
>
> > **W3: [2] I consider this a necessary work to acknowledge, given it is one of the first works discussing the use of attention in MARL**
>
> Thank you for your suggestion. We have now incorporated this method into the related works in Section 2.
>
> > **W4.1: I cannot interpret the error bars in Figure 4—it is too muddled.**
>
> Thank you for your observation. The high variance in the results arises because the metrics we measure during training are near-binary in nature. Specifically, agents either learn to cooperate and achieve scores close to the maximum, or they fail, resulting in a score of zero. Consequently, for methods that mostly succeed or fail errors are very small, but for methods that have failed in a somefraction of the 10 runs, the 95% confidence intervals yield error bars with wide ranges. To address this issue, we have updated Figure 4 to enhance the distinguishability of the error bar edges, ensuring they are easier to interpret.
>
> > **W4.2: Despite the writing overall being clear, the language could be tightened somewhat; e.g. L043: "has to reach its goal" is quite colloquial; also contraction in L497. I recommend combing through the paper and essentially asking each word/phrase to justify itself—and to be as specific as possible, avoiding colloquialisms.**
>
> Thank you for your comment. We revised the submission text to address your concerns.

---

> ### Author Response · Authors · 2024-11-25
> **Response to Reviewer 766H, part 3.**
>
> We appreciate your valuable feedback and have updated our manuscript according to your suggestions:
> - clarified the motivations of the presented baselines, including methods that use communication;
> - added comparison with suggested baselines and extended evaluation on bottleneck environments;
> - included missing references;
> - revised the text to remove colloquialisms.
>
> We hope these revisions clarify our contributions and positively influence your assessment of our work.

---

> > ### Author Response · Authors · 2024-12-02
> > **Request to review detailed response**
> >
> > Dear reviewer 766H,
> >
> > We respectfully request that you kindly revisit our detailed responses, where we have did our best to address your concerns thoroughly. Your feedback has been instrumental in this process, and we hope our response demonstrates our dedication to engaging with the review process and improving the paper.
> >
> > Thank you again for your thoughtful comments, and we hope you find the updates satisfactory.

---

### Author Response · Authors · 2024-11-25

We would like to thank the reviewers for their thoughtful feedback and for recognizing several strengths of our work. All reviewers found an idea and proposed method to be well-explained. Reviewers (766H, r4zr) specifically note that the background and related work are well-discussed. Reviewer fa5a highlights the importance of the challenge and uniqueness of the shared memory approach compared to other communication strategies, mentions that evaluation is rigorous and highlights the potential of SRMT for real-world scenarios in decentralized settings without explicit communication protocols. Reviewer 766H acknowledged that the provided analysis of shared memory is “nice”.

We have carefully considered the reviewers’ questions and comments and have made every effort to address them comprehensively. This involved conducting additional experiments and extending the paper to provide deeper insights. We are confident that these revisions have strengthened the paper and improved its overall quality. Below is a summary of our responses.

**On baselines**

We compare SRMT with MAPF methods that employ different strategies on POGEMA benchmark: centralized with search-based planner (RHCR), without communication (Follower, MATS-LP), cooperative MARL with communication (MAMBA, QPLEX). Follower, MATS-LP, and RHCR are very competitive methods as they are three top-performing from the POGEMA benchmark.
As suggested by reviewers we extended our evaluations with three MARL approaches that employ memory mechanisms (RATE [1], RRNN [2], ATM [3]). On bottleneck environments we found that SRMT overperforms these methods and included results to the revised version of the manuscript.

_Reviewer 766H suggested evaluating RIAL-DIAL, unfortunately this method cannot be directly applied as it relies on sequential decision making, but in the case of MAPF all agents perform actions simultaneously._

**On ablation study**

The bottleneck task results include an ablation study to isolate and demonstrate the role of the proposed shared memory mechanism. In particular, RMT agents (SRMT w\o shared memory) use their individual memory representations locally without sharing them. In the Attention core method (SRMT w\o shared memory and w\o individual memory), the memory is completely removed from the core part of the policy model to test the memoryless architecture. In the Empty core method, the core network is completely removed from the policy model with direct connection of spatial encoder to actor-critic action decoder. The results of these ablation methods are in Figure 3 and show that shared memory is a key component of SRMT, especially in the sparse rewarding scenario, where the task is harder for agents to learn.

------------------------------------
We thank the reviewers for their valuable feedback and the opportunity to update our manuscript. In particular, we have emphasized the results of the ablation study to improve clarity and highlight its significance, and we have incorporated evaluations and discussion of the suggested baselines to provide a more comprehensive comparison.



[1] Cherepanov et al. Recurrent action transformer with memory, 2024. Arxiv:2306.09459.

[2] Santoro et al. Relational Recurrent Neural Networks. In Proceedings of the 32nd International Conference on Neural Information Processing Systems, 2018.

[3] Yang et al. Transformer-based working memory for multiagent reinforcement learning with action parsing. Advances in Neural Information Processing Systems, 35:34874–34886, 2022.

---

### Meta-Review · Area_Chair_4KV5 · 2024-12-29

**Metareview:**

The paper proposes the Shared Recurrent Memory Transformer (SRMT) for improved coordination in decentralized multi-agent pathfinding (MAPF). The core claim is that by pooling and globally broadcasting individual working memories, agents can implicitly exchange information and coordinate actions more effectively without explicit communication protocols. SRMT is evaluated on a bottleneck task and the POGEMA benchmark, where it outperforms various baselines, particularly in sparse reward settings and generalizes well.

The main strengths of the paper are clarity of the presentation, and a comprehensive evaluation including the ablations and memory analysis. The main remaining weaknesses are about novelty, polish and scalability.

Overall, the paper presents a novel approach with promising results. The authors have successfully addressed many of the initial concerns. The revised manuscript has improved the paper, but the original concerns may not be completely addressed.

**Additional Comments On Reviewer Discussion:**

The authors' rebuttal focused on addressing concerns about missing baselines, scalability, and the impact of shared memory. They added new baselines (MAMBA, QPLEX, RATE, RRNN, ATM), performed additional experiments on the bottleneck task and POGEMA benchmark, and provided further analysis on the shared memory mechanism and its role in coordination. The reviewers generally acknowledged these efforts, but some, particularly reviewers r4zr, did not raise their score, maintaining concerns about the paper's novelty and polish. Reviewer fa5a was satisfied with the changes and considered the paper a good contribution.

---

### Decision · Program_Chairs · 2025-01-22

Reject